# Examining the responsiveness of the National Health Insurance Fund to people living with hypertension and diabetes in Kenya: a qualitative study

Robinson Oyando [ID],[1] Vincent Were,[1] Ruth Willis,[2] Hillary Koros,[3] Jemima H Kamano,[4] Violet Naanyu,[4,5] Anthony Etyang,[6] Richard Mugo,[3] Adrianna Murphy,[2] Ellen Nolte [ID],[2] Pablo Perel,[7] Edwine Barasa[1,8]

For numbered affiliations see end of article.

**Correspondence to**
Robinson Oyando;
royando@kemri-wellcome.org

## ABSTRACT

**Objectives** To assess the responsiveness of the National Health Insurance Fund (NHIF) Supa Cover benefit package to the needs of individuals with diabetes and hypertension in Kenya.

**Design, setting and participants** We carried out a qualitative study and collected data using key informant interviews (n=39) and focus group discussions (n=4) in two purposively selected counties in Western Kenya. Study participants were drawn from NHIF officials, county government officials, health facility managers, healthcare workers and individuals with hypertension and diabetes who were enrolled in NHIF. We analysed data using a thematic approach.

**Results** Study participants reported that the NHIF Supa Cover benefit package expanded access to services for people living with hypertension and diabetes. However, the NHIF members and healthcare workers had inadequate awareness of the NHIF service entitlements. The NHIF benefit package inadequately covered the range of services needed by people living with hypertension and diabetes and the benefits package did not prioritise preventive and promotive services. Sometimes patients were discriminated against by healthcare providers who preferred cash-paying patients, and some NHIF-empanelled health facilities had inadequate structural inputs essential for quality of care. Study participants felt that the NHIF premium for the general scheme was unaffordable, and NHIF members faced additional out-of-pocket costs because of additional payments for services not available or covered.

**Conclusion** Whereas NHIF has reduced financial barriers for hypertension and diabetes patients, to enhance its responsiveness to patient needs, NHIF should implement mechanisms to increase benefit package awareness among members and providers. In addition, preventive and promotive services should be included in NHIF's benefits package and mechanisms to monitor and hold contracted providers accountable should be strengthened.

## INTRODUCTION

Non-communicable diseases (NCDs) account for the greatest share of the global disease burden and are one of the significant health challenges of the century.[1 2] Low-income and middle-income countries (LMICs) bear the highest NCD burden.[3] While infectious diseases contribute to the highest share of disease burden in sub-Saharan Africa (SSA), the burden of NCDs is rapidly increasing and is expected to grow.[4] In Kenya, NCDs account for 50% of inpatient admissions and 55% of hospital deaths.[5] Responsive health systems provide high-quality health services equitably and efficiently to the population in need without the risk of catastrophic or impoverishing healthcare expenditures.[6 7] Evidence suggests that most NCD patients in LMICs do not have equal access to NCD services such as screening and treatment.[8–13] Among other things, this is due to lack of access due to financial barriers and inadequate capacity of health facilities to offer NCD care.[8 12–14]

One of the mechanisms to enable financial access to NCD services is designing and implementing prepayment health financing mechanisms from tax or compulsory insurance contributions that provide financial risk protection to households and individuals living with NCDs and enhance access

to quality NCD services.[15 16] Some evidence suggests that having health insurance is beneficial in enabling access to NCDs services.[17–19] Nevertheless, health insurance coverage in SSA remains low, with only 4 out of 36 SSA countries having health insurance coverage of any type exceeding 20%.[20] In addition, tax-based prepayment mechanisms are preferred as an equitable health financing mechanism in SSA given the context of a large informal sector with a low contributory capacity.[21]

## STRENGTHS AND LIMITATIONS OF THIS STUDY

⇒ This study used in-depth interviews and focus group discussions to comprehensively examine how responsive the national health insurance fund is to the healthcare needs of patients with hypertension and diabetes.

⇒ The study recruited participants from both urban and rural sites and health staff from a range of cadres and thus provides evidence from a wide range of respondents with varied experiences.

Kenya has implemented several health financing initiatives.[22–25] One such initiative is the policy decision to use the National Health Insurance Fund (NHIF) as one of the key prepayment health financing mechanisms.[26] NHIF is a public insurer in Kenya that was established in 1966 to provide mandatory health insurance to all Kenyans.[22] However, national NHIF coverage remains low, at 24%.[27] In the past 11 years, several reforms have been introduced in NHIF in an effort to enhance its capacity to deliver universal health coverage.[22] Among these reforms is the broadening of the benefit package ('Supa Cover') of its general scheme (that targets all Kenyans) to include outpatient and specialised services for NCDs.[22]

There is limited evidence about the degree to which the expanded NHIF national scheme is addressing the needs of people with hypertension and diabetes—two major NCDs in Kenya. This study aims to generate evidence that would be relevant in informing policymakers and other stakeholders on how the configuration of health insurance reforms can be tailored to address the needs of NCD patients and thus contribute to the achievement of international benchmarks for NCDs spelt out in Sustainable Development Goal 3.4.

## METHODS
### Study setting
Kenya is an LMIC located in the East Africa region.[28] There are two levels of administration in Kenya's health system: national and county.[29] The national government (Ministry of Health) is responsible for policy and regulation in the health system while the 47 county governments are responsible for service delivery.[30] The Kenya health system is pluralistic, with an almost equal share of service provision by the public and private providers.[31]

Kenya's health system has a four-tiered structure. Tier 1 is at the community level where community-based healthcare demand creation activities are undertaken as stipulated in the community health strategy; Tier 2 is at the primary care level where preventive and promotive services are provided in dispensaries and health centres; Tier 3 consists of both primary and secondary referral hospitals; and Tier 4 consists of tertiary referral hospitals under the direct management of the Ministry of Health (MoH).[32]

The sources of health financing in Kenya include out-of-pocket (OOP) payments by households, government (national and county) tax revenues, donors and member contributions from NHIF and private insurance companies.[33 34] For instance, in the financial year (FY) 2015/2016, current health expenditure (CHE) as a percentage of gross domestic product was 5.2%, CHE per capita in US$ was 78.6 while the proportion of CHE financed through OOP payments was 26.1%.[33] In addition, the total health expenditure (THE) on NCDs as a proportion of THE was 11% in FY 2017/2018.[35]

### Study sites
This study was carried out in Busia and Trans Nzoia counties in Western Kenya, where Moi University, Moi Teaching and Referral Hospital through AMPATH (Academic Model Providing Access to Healthcare) have partnered for several years with county governments to strengthen health systems across various care levels.[36]

This study is part of a larger study that seeks to inform and support the scale-up of the Primary Health Integrated Care Project for Chronic Conditions (PIC4C) model for integrated management of people with hypertension, diabetes and breast and cervical cancers in Kenya.[37] Alongside other work packages in the larger study, our work package had three overarching objectives: (1) to measure the effectiveness of the NHIF national scheme benefit package to provide financial risk protection to individuals with hypertension/diabetes; (2) to examine the extent to which the NHIF national scheme benefit package is responsive to the needs of individuals with hypertension/diabetes; and (3) to examine how the provider incentives generated by provider payment arrangements of the NHIF national scheme benefit package influence equity, efficiency and quality of care. Our engagement with policymakers on the findings of the entire study included presentation of findings from both the quantitative (objective 1) and qualitative (objectives 2 and 3) work. However, the focus of this paper is on objective 2.

Table 1 outlines the key characteristics of the two counties. Launched in 2018, the PIC4C model was implemented by a partnership between AMPATH/Moi, the MoH and the World Bank and included (1) early case finding of people with hypertension, diabetes, cervical/breast cancer at service level 1; (2) structured referral to service providers at level 2 for confirmation of diagnosis and treatment initiation or referral to level 3 or 4 using structured protocols; (3) initiation of treatment using structured treatment protocols and decision support tools at levels 2, 3 and 4; (4) retention of patients in care supported by ongoing training of health workers at all care levels; (5) monitoring and evaluation supported by a health information system; and (6) advising patients in care to enrol in the NHIF national scheme for sustainable health financing.[37]

### Study conceptual framework
This study's conceptual framework (figure 1) assumes that for the NHIF to be effective in providing financial risk protection to patients with hypertension/diabetes and their households while enhancing access to quality

**Table 1** Characteristics of the study setting

| Characteristics | Busia county | Trans Nzoia county |
|---|---|---|
| Population[51] | 893 681 | 990 341 |
| % Adults (18+ years)[51] | 52% | 53% |
| % Females[51] | 52% | 51% |
| Life expectancy at birth[37] | 47 years | 60.5 years |
| Poverty level[37] | 64% | 50% |
| Estimated hypertension prevalence[52] | 23.1% | 33.6% |
| Estimated diabetes prevalence[37] | 1.5% | 2% |
| NHIF coverage[37] | 31% | 20% |
| Number of health facilities[53] | 184 community units served by 47 dispensaries, 12 health centres, 5 subcounty and 1 county hospital. | 198 community units served by 38 dispensaries, 8 health centres, 6 subcounty health facilities and 1 county hospital. |
| Number of patients with selected NCDs currently being serviced in AMPATH sites[37] | 6060 patients with hypertension, 1113 with diabetes and 200 with cervical cancer. | 4375 patients with hypertension, and 2089 with diabetes. |
| Number of facilities where the PIC4C pilot was implemented[37] | 40 facilities (5 subcounty hospitals, 12 health centres and 23 dispensaries). | *33* facilities (6 subcounty hospitals, 8 health centres and 19 dispensaries). |

AMPATH, Academic Model Providing Access to Healthcare; NCD, non-communicable disease ; NHIF, National Health Insurance Fund; PIC4C, Primary Health Integrated Care Project for Chronic Conditions .

health services, it is imperative that (1) the benefit package represents services that patients need, (2) that patients actually access the entitlements from the benefit package and (3) the NHIF reimbursements to healthcare providers adequately covers the fees providers charge patients for services (depth of cover). Therefore, following the study's conceptual framework, responsiveness in this study is conceptualised to mean that the NHIF Supa Cover meets the needs of diabetes and hypertension patients in each of the three elements.

## Study design and data collection

We conducted a qualitative study where we interviewed purposively selected participants. We conducted four focus group discussions (FGDs) with patients with diabetes and hypertension in Busia (one rural and one urban) and Trans Nzoia (one rural and one urban) counties who were enrolled in the NHIF national scheme. FGD participants were selected from an existing PIC4C scale-up study database (matched cohort wave 2).[37] FGD

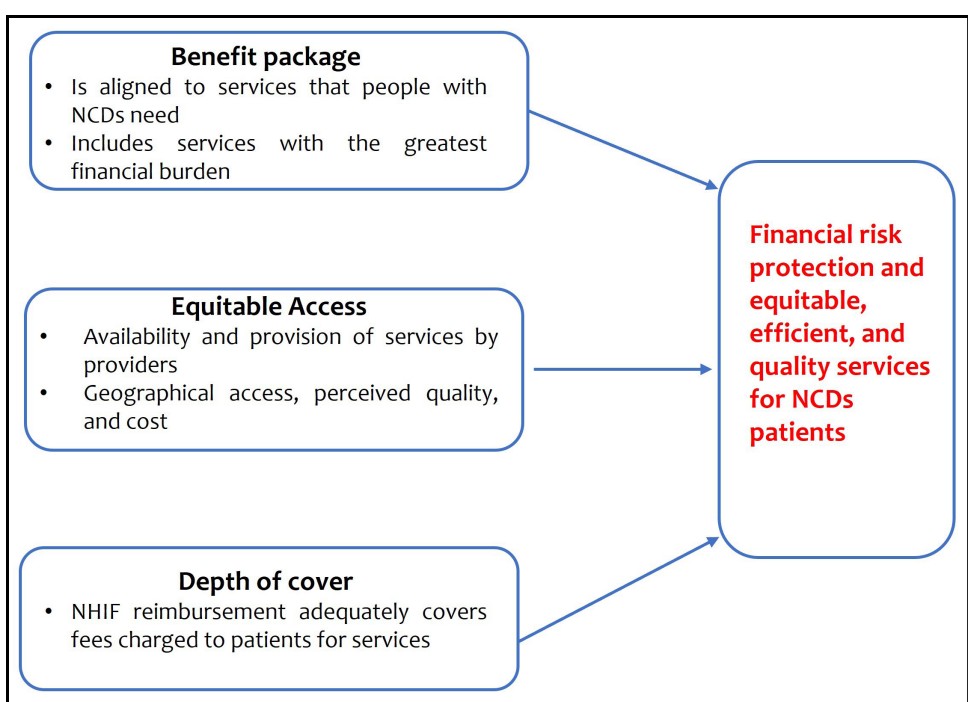

**Figure 1** Study conceptual framework. NCDs, non-communicable diseases; NHIF, National Health Insurance Fund.

**Table 2** Summary of respondents

| County | Respondent | No. interviewed |
|---|---|---|
| Busia | County officials | 4 |
| | NHIF branch officers | 1 |
| | Facility managers | 6 |
| | Frontline healthcare workers | 10 |
| | NCD patient FGDs | 2 (1 urban (7 participants); 1 rural (6 participants)) |
| Trans Nzoia | County officials | 3 |
| | NHIF branch officers | 2 |
| | Facility managers | 6 |
| | Frontline healthcare workers | 7 |
| | NCD patient FGDs | 2 (1 urban (7 participants); 1 rural (5 participants)) |

FGDs, focus group discussions; NCD, non-communicable disease; NHIF, National Health Insurance Fund.

participants were eligible if (1) they were actively enrolled in NHIF at the time of recruitment, (2) they had either diabetes or hypertension or comorbid (both diabetes and hypertension), (3) were more than 18 years of age and (4) if they consented to participate in the study. Patients were purposively selected to ensure a range of age, gender, condition/s and attendance at urban/rural facility. Eligible participants were invited to the FGD sessions by telephone. FGD sessions took place at the nearest public health facility and at a time convenient to the participants. In-depth interviews (IDI) participants were purposively selected to include a range of cadres in both counties. IDIs were held with county health officials (n=7), county NHIF branch officers (n=3), health facility managers (n=12) and frontline healthcare workers (n=17) composed of medical officers, pharmacists, nursing officers and clinical officers drawn from public providers (health centres (n=2), subcounty hospitals (n=3) and county hospital (n=1)) and private providers (faith-based facility (n=1) and health centre (n=1)). All study participants consented and signed informed consent forms before participating in the interview sessions. FGD and IDI sessions were facilitated by two trained qualitative researchers. The FGDs were facilitated in Kiswahili while the IDIs were conducted in English. With consent from the participants, the sessions were audio recorded and complimentary notes were taken. FGDs lasted for an average of 48 min while the interviews lasted for an average of 35 min. Table 2 outlines the number of respondents interviewed by their respective categories.

The FGDs and IDIs were conducted using semi-structured topic guides. The development of the questions in the discussion and interview guides was informed by the study's conceptual framework (figure 1). The questions focused on hypertension/diabetes services that NHIF pays for, adequacy of the range of services covered by NHIF, participants' experience with service provision, quality of care, prioritisation of preventive and promotive services in the NHIF Supa Cover scheme, NHIF provider payment mechanisms, strengths and weaknesses of the NHIF national scheme. At the end of each session, the facilitator and note taker held debrief sessions which facilitated improving the interviewing process and revision of some questions to enhance clarity and inclusion of emerging themes that needed further exploration. Data collection was discontinued when data saturation was attained (ie, when no new information was forthcoming). The total sample size (39 interviews and 4 FGDs) meets the recommendation for sample size (9–17 for interviews and 4–8 for FGDs) for qualitative studies to achieve saturation.[38] Data were collected between June and July 2021.

### Data management and analysis

Qualitative FGDs and IDIs audio recordings were transcribed verbatim and translated to English where necessary. To validate and familiarise ourselves with the transcribed transcripts, we read and reread the transcripts and ensured correctness by relistening to sections of the audio recordings. Validated transcripts were imported into NVivo QSR V.12 for coding and analysis. We used a thematic approach to analyse the qualitative data. Thematic analysis involved familiarisation with the data through reading and rereading the transcripts to identify emerging ideas, meanings, patterns and potential themes.[39] In the first stage of analysis, two authors (RO and RW) working independently coded the first few transcripts to identify key themes and developed the initial coding framework. This involved the generation of initial codes, grouping developed codes into descriptive themes by identifying matching patterns. Thereafter, a discussion among three authors (RO, RW and EB) was held where individual codes were compared, and a final coding framework was arrived at by consensus. Whereas the coding of the transcripts was guided by the study's conceptual framework, as data coding progressed iteratively, the coding framework was updated based on the emerging codes. In the next step, RO charted the data. During charting, emerging themes are identified through a process where coded data is reorganised. In this stage, RO summarised the ideas from each code with illustrative quotes from the data. EB checked and refined coding charts developed by RO. In the final phase, summaries from each thematic heading were critically interpreted and synthesised to identify key policy-relevant findings.

### Trustworthiness

We built trustworthiness in the study findings by interviewing multiple participants to identify different perspectives (data source triangulation), using different methods of data collection (method triangulation), iterative questioning through rephrasing of questions, use of probes, holding peer debriefing sessions with the study team and member checking where we shared preliminary

findings with participants and asked them to verify the resonance of our findings with their perspectives.[40] We ensured analytical rigour by using multiple coders; our coding team included an experienced sociologist (RW) as well as experienced health systems researchers (RO and EB).[40]

### Reflexivity
All authors have conducted health systems research on NCDs across different LMICs. Specifically, three authors have previously conducted research on the economic burden of hypertension and diabetes in Kenya which influenced their interest in the study topic and methodology including sampling of participants, data collection methods and data analysis.

### Patient and public involvement statement
The roll out of the PIC4C model in the two counties was accomplished by involving village chiefs and elders in the process. Patients and members of the public were not involved in the development of the proposal, or the design of the research presented here.

### RESULTS
We conducted a total of 39 IDIs (17 men and 22 women) and 4 FGD sessions with 28 participants (15 women and 13 men). Six FGD participants had diabetes, 13 had hypertension and 9 had both diabetes and hypertension. The results are presented under five key thematic areas: access to hypertension/diabetes services, awareness of hypertension/diabetes service entitlements in the NHIF Supa Cover scheme, adequacy of the range of hypertension/diabetes services covered by NHIF, provider payment mechanisms for hypertension/diabetes services and affordability of NHIF Supa Cover scheme. Summaries of key findings are provided under each thematic area.

### Access to hypertension/diabetes services
**The NHIF benefit package expanded access to services for people living with hypertension/diabetes by reducing financial barriers**
Several participants reported that the NHIF package had facilitated access to a range of hypertension/diabetes care services that they would otherwise have had to pay for OOP. Services that were reported to be covered included inpatient care and referral to NHIF accredited healthcare facilities, and medications. For example, one patient noted:

> When I have the NHIF, I have hope that I can go to the hospital even if I don't have any money and I can get medication compared to the one who does not have the NHIF. Rural FGD Participant, County A

One healthcare worker specifically related NHIF coverage to increased clinic attendance:

> The strength of NHIF super cover is that it has alleviated the suffering of these patients. There are patients who had stopped coming to the clinic because

of financial challenges, but with the NHIF cover, we found that almost all our patients are always coming for the clinic…those who are covered are always coming for the clinic without fail. Healthcare worker, Subcounty Hospital, County A

Being enrolled in NHIF was reported to have beneficial implications for patients' wider financial situations, for example, avoiding the need to borrow money or sell assets to pay for healthcare services.

> It prevents you from borrowing and reduces stress for you and even respect comes. It prevents you from selling the farm, animals and any other thing. Rural FGD Participant, County B

Further, financial barriers were reduced for all population groups because the NHIF did not limit the enrolment of individuals based on age or pre-existing NCD condition, unlike private health insurance.

### Awareness of hypertension/diabetes service entitlements
**Some NHIF members and healthcare workers had inadequate awareness of the NHIF service entitlements**
While some study respondents showed awareness of the hypertension/diabetes care services that are covered by the NHIF Supa Cover scheme, others indicated that they were not adequately aware of service entitlements under the scheme.

> If you were to get a patient who needed radiotherapy or chemotherapy in my facility, we don't offer those services. I don't know if NHIF caters for that. I don't know to what extent…it does not cover some services such as chemotherapy or radiotherapy. I don't know if it caters for them, am not sure. I hope it does. Healthcare worker, Subcounty Hospital, County A

Inadequate awareness of hypertension/diabetes service entitlements in the Supa Cover scheme resulted in some patients being wrongfully turned away by healthcare workers.

> I asked if the NHIF card can be used to do X-ray and whoever was there told me that it cannot, you have to go outside the hospital. I didn't go and that is when I realised that the NHIF can do X-ray. That is when I realised that there are others who get services while others do not. The last time there was a nurse who came and asked on my behalf and she was told that it was working, and I got the services. Urban FGD Participant, County B

### Adequacy of the range of hypertension/diabetes services
**The NHIF benefit package inadequately covered the range of services needed by people living with hypertension and diabetes**
While the cover catered for outpatient consultation and inpatient admissions, there was a strong perception among most NHIF members and healthcare workers that the NHIF Supa Cover package did not adequately cover the range of diagnostic, monitoring and medicines that

they needed and did not explicitly cover hypertension/diabetes medicines. Several participants reported incidents of relevant services having to be covered OOP by service users themselves.

> One day I came here and they sent me to go get an X-ray. When I got there, I was told that they cannot cover for that expense. "Go out". When I got out there, I asked and was told KES 5200/= to get the X-ray. I was perplexed. Urban FGD, County A

> First thing urinalysis, HbA1c [glycated haemoglobin] and UEC [urea, electrolytes and creatinine]. UEC will guide us whether to start with insulin oral or metformin alone. Then afterwards you need to follow up on this patient after 3 months you need HbA1c, but you find most of our clients mostly those relying on insurances, they're not able to pay so I can comfortably say they are not adequately covered using NHIF. Medical Officer, Private Hospital, County A

### The NHIF benefit package did not prioritise preventive and promotive services for hypertension/diabetes

Some study respondents felt that the NHIF Supa Cover does not prioritise preventive and promotive services such as screening for NCDs but instead prioritises curative care after a patient is diagnosed with hypertension or diabetes. It was further felt that the NHIF prioritised secondary and admission care in hospitals rather than primary care services at lower-level facilities.

> The Supa Cover mostly deals with curative aspects of treatment. So, it doesn't deal with preventive aspects of treatment. So, it waits for a client to fall sick, then they can get services. There is no such thing as a preventive service, maybe going for annual check-up unless of course, you are covered under the civil servant type of cover where you have access to one check-up per year. Facility Manager, Subcounty Hospital, County A

> We were told that NHIF is able to cater for health checks, routine screening. And unfortunately, that one is not being done for everybody; it is not for the supa cover, it is for these other packages. County Official, County B

### Provider payment mechanisms for hypertension/diabetes services
#### The design of the outpatient capitation payment mechanism reduced geographical access to services

The capitation payment mechanism required patients to be registered to a specific health facility for outpatient services. Some participants mentioned that this limited access to outpatient services when patients travelled far from their registered facility, and hence compromised continuity of care.

> The government should look into that… [so that] if you have NHIF, you can receive treatment anywhere.

> It is not a must that one chooses a hospital to receive treatment. Rural FGD Participant, County A

### Quality of hypertension/diabetes care

#### NHIF patients were sometimes discriminated against by healthcare providers

Some study participants felt that cash-paying patients received more attention and care from healthcare workers than patients with hypertension/diabetes enrolled in NHIF. For instance, respondents living with hypertension/diabetes observed that healthcare workers did not give them the support they required when in a health facility once they mentioned that they had NHIF cards.

> When you walk in the corridors and meet a doctor and you tell them you have NHIF card, they look at you differently. It is like people see you as desperate because you are mentioning that card. I think they prefer cash because if you say you have the card, they don't want to assist you. Urban FGD Participant, County B

> There are times when you waste time walking around the hospital with no help until you decide to use cash instead of wasting the whole day. Sometimes they have a negative attitude if you talk to them about the card. Urban FGD Participant, County B

However, other participants had dissenting views as they reported that there was no difference in how healthcare workers handled patients with hypertension/diabetes enrolled in NHIF and those who are not.

> No favouritism is there because one is using NHIF and the other one is using cash. Rural FGD Participant, County B

Some patients related how they were sometimes denied the healthcare services they needed because they had an NHIF card. These patients were instead asked to seek services from private providers.

> I was just given nifedipine alone for blood pressure. I had my fellow patient who was given all the drugs because she paid cash. She was given all drugs. I left there asking myself what the card had done. I was discharged and told to go and buy drugs. I was given only one drug. Urban FGD Participant, County B

#### Some NHIF-empanelled health facilities had an inadequate structural quality of care

This was characterised by overcrowding at the clinics, erratic supply of hypertension/diabetes medicines, diagnostic supplies and equipment. Some participants reported that they had to return on another clinical day or seek services in other (often private) health facilities and incur OOP.

> Again, after you've come for clinic, there is a schedule, and patients are many. You are told that time is up

for receiving your patient book. Yet you got delayed at the laboratory. I have seen others being returned. Their book gets rejected. Urban FGD Participant, County A

The challenge that is there, it is like there are some tests that they don't do. Like today I did ECG [electrocardiogram], to know the condition of my heart because the other day I came to the hospital. I told the doctor that I had pain that I could not understand. Then I was told to come and do ECG. The machine for ECG was spoilt and because I wanted the service, I had to go outside the hospital, and I paid. Urban FGD Participant, County B

Some patients and healthcare workers reported that public health facilities often did not provide inpatient diets that were suitable for patients with hypertension and diabetes. The meals they were served during inpatient admission in public facilities were contrary to the dietary advice they were given to manage their conditions.

If they prepare *uji* (porridge), it has a lot of sugar. And you have been told not to take sugar. Who will prepare yours now? If they knew that, they would set apart a ward. Rural FGD Participant, County A

### Affordability and financial risk protection
#### Study respondents felt that the NHIF premiums for the general scheme were unaffordable
The KES 500 per month premium was deemed to be beyond reach by some respondents, especially in rural areas. As a result, people living with hypertension/diabetes could not keep up with the premium payments and regularly defaulted.

The NHIF should reduce the premiums since the people in the rural areas see KES 500 as a lot of cash and fail to register or remain active. Rural FGD Participant, County B

Furthermore, individuals from the informal sector (whose membership was voluntary) who needed to undergo costly or complicated procedures were required to pay their monthly premium subscription at least 1 year in advance before NHIF authorised them to undergo any procedure. This was deemed unaffordable by many respondents.

NHIF members from the informal sector are normally required to pay premiums at least more than 2 years in advance, depending on the cost of the procedure that the person is required to do. They are required to pay premiums for 2 years or more before the preauthorisation can be approved. Healthcare worker, FBO, County A

If you're a NHIF contributor and self-employed, you pay the KES 500 package every month. You cannot get the surgery done unless you get the pre-authorised form, and you should have paid premiums upfront

for 1 year for the surgery to be approved. You will find that most of these patients are old, some of them are unemployed so while they can pay KES 500 per month, it becomes a very big issue to pay for a year or more. County Official, County A

#### NHIF members faced additional out-of-pocket costs because of administration charges, input supply shortages, balance billing and payments for services not covered
Due to supply-side inefficiencies, some patients reported that they were required to make OOP to purchase consumables like gloves or photocopying fees for receipts.

Everyone paying for NHIF is struggling. Like the other day on Wednesday. The day has come, we get there, KES 10/= for photocopy and KES 50/= for gloves. Rural FGD Participant, County B

Further, the reimbursements NHIF paid providers for inpatient and outpatient services were deemed insufficient to cover the cost of offering the services. Hence, patients had to make additional payments to bridge the gap in service delivery costs.

Per day NHIF payment for inpatient is KES 1500, while outpatient capitation is KES 1200 so they can actually surpass. In a day the bill can actually go beyond that because we are looking at for example cost of insulin, the cost of the drugs they are getting…So, in as much as NHIF would pay the KES 1200, now the patient has to top up, yes. Facility Manager, County Referral Hospital, County A

Services that NHIF did not cover included dental services, eye examination, physiotherapy, X-ray and laboratory tests such as glycated haemoglobin test, liver or kidney function tests, urea, electrolyte, creatinine, among others.

…when you've been sent to physiotherapy. They will tell you to add a certain amount of money that they will tell you. They say that the card does not cover that. Urban FGD Participant, County A

Some tests like for the diabetics, if you want to do an HbA1c, they have to pay cash. If they have to do a thyroid function test, they have to pay cash. So, there are some tests, even PSA [prostate-specific antigen], for the men with the NCD, they have to pay in cash. So, I have really never understood why, whereas when they go to NHIF they are told it is fully covered by NHIF. Healthcare worker, County Referral Hospital, County B

As a result of these additional costs, some patients were reported to skip clinical visits hence compromising continuity of care.

First of all, he has issues with the fare to the facility. Reaching the facility, he has to be sent to go and do a RBS [random blood sugar] or a FBS [fasting blood sugar]. That is an extra cost. From there he has been

told to go buy drugs for 3 months and he is unable to even buy worth 1 week. So, he might end up absconding from clinic or just stopping the care completely. County Official, County A

## DISCUSSION

This study aimed to assess the responsiveness of the NHIF general scheme to the needs of individuals with hypertension and/or diabetes. Our findings are consistent with international evidence that having health insurance can enhance access to NCDs services.[17–19] Study respondents reported that having NHIF was critical in improving access to NCDs services. However, purchaser (NHIF) and provider (healthcare facilities) factors continue to impair the responsiveness of the NHIF to the needs of individuals living with hypertension/diabetes. On the purchaser side, documented challenges included inadequate awareness about service entitlements in the NHIF Supa Cover package. Inadequate awareness of hypertension/diabetes service entitlements by healthcare workers and patients resulted in denial of care as patients would sometimes be turned away. A similar finding has been reported in a previous study.[41] In addition, respondents felt that the NHIF Supa Cover did not prioritise preventive and promotive services for NCDs but rather focused on curative services at secondary hospitals, that are not accessible to the rural poor. Nevertheless, targeted NCD screening is cost-effective[42–44] and there is evidence that the poor need NCD screening services more than the rich in Kenya.[13]

Unaffordability of the NHIF Supa Cover monthly premiums and the penalties of defaulting payment were among the key weaknesses identified by study participants. This was specifically a challenge for enrollees from the informal sector, a finding that has been reported in previous studies conducted in Ghana and Kenya.[41 45 46] Furthermore, patients with hypertension/diabetes enrolled in the NHIF Supa Cover from the informal sector were required to pay their monthly premium subscription at least 1 year in advance before NHIF provided authorisation for them to undergo costly or complex procedures. This presents a substantial financial barrier to patients and goes counter to the role of health insurance as a mechanism for pooling risks to avert the need to make large OOP payments when a health event occurs. Despite the NHIF providing some financial protection, OOP payments persisted. The unavailability of NCD medicines in public facilities has been corroborated by the findings of a previous study conducted in eight Kenyan counties.[47] Similarly, there is evidence of inequities in hypertension treatment in Kenya, with the poor being disproportionately disadvantaged.[13]

Several purchaser and provider factors compromised physical access to NHIF-contracted facilities and quality of care for hypertension/diabetes services. First, except for inpatient admission, capitation-based payments by NHIF limited access to hypertension/diabetes services to only one NHIF-accredited provider. As such, patients had to pay OOP to access hypertension/diabetes services from other providers if the services they needed were unavailable from the providers where they were capitated. Second, infrastructural gaps hindered access and compromised the quality of hypertension/diabetes services as patients were sometimes turned away and the required laboratory supplies and equipment were often broken down or lacking in most public facilities contracted by NHIF.

Whereas this study did not formally assess provider factors that influence service experience by individuals enrolled in the NHIF Supa Cover, there were provider factors that influenced respondents' experience of the Supa Cover benefit package. For instance, consistent with available literature,[23 48] some patients felt discriminated against by healthcare workers because they were enrolled in NHIF. Healthcare workers' negative attitudes towards NHIF enrollees is likely to lead to attrition.[41] This finding could be explained by NHIF's capitation-based provider payment mechanisms that are low and unpredictable.[49]

This study had limitations. First, while we aimed to elicit experiences from patients with hypertension/diabetes who were enrolled in the NHIF Supa Cover scheme at the time of the interview, including patients who had discontinued membership from NHIF could have yielded richer findings on NHIF responsiveness. Nevertheless, the findings from the quantitative component of our study[50] that has included households that were enrolled into NHIF, discontinued their enrolment, or were not completely enrolled complements the findings of this qualitative work. Second, the study was limited to two counties, and a larger study across more counties may identify a greater range of experiences. However, the fact that the NHIF is a national agency, and the implementation of its programmes, for example, the benefit package or the premium contributions is standardised across geographies, gives confidence that observations in two countries are likely comparable to those in other counties.

Based on our study findings, we make the following policy recommendations. First, the NHIF should enhance its engagement and communication with its members living with hypertension/diabetes, health facilities and county managers to improve awareness about the entitlements in the NHIF benefit package. Second, the NHIF should review its benefit package to align with the needs of people living with hypertension/diabetes. This includes considering the inclusion of key diagnostic, and monitoring services for hypertension/diabetes, and expanding priority to include preventive and promotive services which are more cost-effective. Third, the NHIF should review the design of the outpatient capitation model to make provisions for portability. That is, the capitation mechanism should provide for individuals to access services in facilities that they are not capitated to in instances when this is necessary (such as when they travel or in case of an emergency). Fourth, the NHIF and county

governments should strengthen monitoring and accountability of health facilities to address undesired provider behaviour such as discrimination. Fifth, to address the structural quality issues of health facilities, the NHIF and county governments should enhance monitoring. Further, the NHIF should enhance its facility selection mechanism to assure the selection of health facilities with the structural capacity to deliver high-quality services, while the county government and health facilities should prioritise investments to improve the structural capacity of public health facilities. Health facility managers also have a role in implementing internal mechanism to prevent and address undesired provider behaviour through monitoring and sanctions. Sixth, the national government should consider providing full or partial subsidies to enhance the affordability of the NHIF premiums in the informal sector. This should be accompanied by a review of the additional administrative charges to enrol with the NHIF. Last, the NHIF should review its pre-authorisation requirements to remove financial barriers. Specifically, the requirement for NHIF members from the informal sector to make long-term premium payments before accessing expensive procedures goes counter to the intention to provide financial risk protection through pooling.

## Conclusion

While the NHIF has reduced financial barriers and enhanced access to care for people living with hypertension and diabetes in Kenya, challenges persist that compromise NHIF's responsiveness to the needs of these patients. As the government of Kenya embarks on the national roll out its national UHC programme, a window of opportunity exists to enhance NHIF's responsiveness to the needs of patients living with hypertension/diabetes. The responsibility to do this is shared among key actors like the NHIF, the national and county governments, as well as health facilities.

**Author affiliations**
[1]Health Economics Research Unit (HERU), KEMRI-Wellcome Trust Research Programme, Nairobi, Kenya
[2]Department of Health Service Research and Policy, London School of Hygiene and Tropical Medicine Faculty of Public Health and Policy, London, UK
[3]Academic Model Providing Access to Healthcare, Eldoret, Kenya
[4]Department of Medicine, School of Medicine, College of Health Sciences, Moi University, Eldoret, Kenya
[5]School of Arts and Social Sciences, Moi University, Eldoret, Kenya
[6]Department of Epidemiology and Demography, KEMRI-Wellcome Trust Research Programme, Kilifi, Kenya
[7]Department of Non-Communicable Disease Epidemiology, London School of Hygiene and Tropical Medicine, London, UK
[8]Center for Tropical Medicine and Global Health, University of Oxford, Oxford, 01540, UK

**Acknowledgements** The authors wish to acknowledge the county health officials from Busia and Trans Nzoia, research assistants and study participants for enabling the successful implementation of this research study.

**Contributors** PP, JHK, EN, VN, AM, AE and EB conceptualised the study and obtained funding. EB, VW, RO, RW and HK contributed to the methodological development and study design. HK, RM, RO and VW supported participant recruitment and data collection. EB, RW and RO analysed and interpreted the data. RO prepared the first draft manuscript that was subsequently reviewed and revised by all the authors. All authors read and approved the final manuscript. RO accepts full responsibility for the finished work and/or the conduct of the study, has access to the data, and made the decision to publish.

**Funding** This work was supported by the UK Medical Research Council grant number MR/T023538/1. RO is funded by a Wellcome Trust International Masters award (#214622), and additional funds from a Wellcome Trust core grant awarded to the KEMRI-Wellcome Trust Research Program (#092654) supported this work.

**Competing interests** None declared.

**Patient and public involvement** Patients and/or the public were involved in the design, or conduct, or reporting, or dissemination plans of this research. Refer to the Methods section for further details.

**Patient consent for publication** Not applicable.

**Ethics approval** This study was reviewed and approved by the Institutional Ethics and Research Committee of Moi University (IREC 0003586) and the London School of Hygiene and Tropical Medicine (17940).

**Provenance and peer review** Not commissioned; externally peer reviewed.

**Data availability statement** All data relevant to the study are included in the article or uploaded as supplementary information.

**ORCID iDs**
Robinson Oyando http://orcid.org/0000-0001-8768-6014
Ellen Nolte http://orcid.org/0000-0002-2289-117X

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
