## [Reviewer comments · BMJ Open]

This paper was submitted to a another journal from BMJ but declined for publication following peer review. The authors addressed the reviewers' comments and submitted the revised paper to BMJ Open. The paper was subsequently accepted for publication at BMJ Open.

ARTICLE DETAILS

TITLE (PROVISIONAL)	Examining the Responsiveness of the National Health Insurance Fund (NHIF) to People Living with Hypertension and Diabetes in Kenya: A qualitative study
AUTHORS	Oyando, Robinson; Were, Vincent; Willis, Ruth; Koros, Hillary; Kamano, Jemima; Naanyu, Violet; Etyang, Anthony; Mugo, Richard; Murphy, Adrianna; Nolte, Ellen; Perel, Pablo; Barasa, Edwine

VERSION 1 – REVIEW

REVIEWER	Dalinjong, Philip Ayizem Navrongo Health Research Centre, Ghana Health Service
REVIEW RETURNED	01-Dec-2022

GENERAL COMMENTS	The manuscript is well written and interesting to read. It is a great contribution to the field given the increasing burden of non-communicable diseases in developing countries. It's also important in the era of universal health coverage. Currently, the manuscript requires proofreading. Authors should proofread to correct sentence structure in terms of punctuation, grammar and omissions.
--

REVIEWER	Chaturvedi, Abhishek Virginia Commonwealth University
REVIEW RETURNED	09-Jan-2023

GENERAL COMMENTS	In this study by Oyando et al., authors examined responsiveness of the National Health Insurance Fund (NHIF) to needs of individuals living with hypertension and diabetes in Kenya and had multiple findings and conclusions after focused grouped discussed and in-depth interviews. The biggest weakness in the study is its small sample size (only 2 counties and handful of patients were included) and qualitative study design. This limits the generalizability of these findings to the participants who were not interviewed and falls short of the aim to assess impact of NHIF and areas of improvement. Moreover, authors' intent was to "generate evidence that would be relevant in informing policymakers and other stakeholders on how the configuration of health insurance reforms can be tailored to address the needs of NCD patients". This goal can not reliably be
---

achieved without an objective study where there are specific questions and responses to be picked by the respondents. For example, authors need to show policymakers whether implementation of NHIF increased the actual number of screening visits or follow up visits, if there is change in proportion of patients with controlled diabetes or blood pressures etc. If there is negative impact due to cost issues or lack of understanding, this can be captured by surveying the participants regarding the reasons for not following up or not taking the medications. In summary, authors can use this qualitative data to build objective questionnaire to be circulated to eligible patients. This way they would have more participants and objective data as to the positive/negative impact of NHIF on the screening/control/treatment of NCDs in the population.

The title and conclusion of the manuscript mentions "diabetes and hypertension" but majority of other text in the manuscript mentions "NCD". Even the responses from the respondents talk about issues with chemotherapy/radiotherapy/X-ray/PSA etc that are issues not closely related to diabetes/hypertension screening/management/follow up. This is again likely because respondents did not have a specific set of questions-answers to stick to which again makes it harder to apply the study findings to a broader population.

Authors conclude, "While the NHIF has reduced financial barriers and enhanced access to care for people living with hypertension and diabetes in Kenya, challenges persist that compromise NHIF's responsiveness to the needs of these patients". Again this claim is only based on responses from few participants. What if an objective study later finds that majority patients believe that due to non-affordability of NHIF premiums, they actually have reduced access to care?

Introduction section is way too long before it gets to the main reasons of why this study was conducted. Consider it shortening.

Authors report that NHIF coverage remains low at 16% but that data comes from a study in 2017. Do authors have more updated info on the current %age of population covered by NHIF?

Study design and data collection, para 1: "FGD participants were selected from an existing PIC4-C scale-up study database". How were the participants selected. Based on eligibility criteria, there must be more eligible participants in the county. How many were actually screened for the study?

Study design and data collection, para 2: "Data collection was discontinued when data saturation was attained". Can authors explain what they mean by that?

Citation 23: The link lead to a website and not actual content/study/publication, so I am unable to verify if the data mentioned here is correct.

Table 1: Need to cite the source of data for estimated hypertension and diabetes in respective counties.

	There are many incorrect usage of punctuations especially in the respondent answers but also in other parts of the manuscript. Since these were deduced from the interviews and not really written by the patients themselves, would recommend making sure sentences are coherent with correct punctuation usage.
--	--

VERSION 1 – AUTHOR RESPONSE

Reviewer 1 Comment(s)	Response
Authors should proofread to correct sentence structure in terms of punctuation, grammar, and omissions.	We thank the reviewer for this comment. We have proofread the entire manuscript and made changes made where necessary.
Reviewer 2 Comment(s)	Response
The biggest weakness in the study is its small sample size (only 2 counties and handful of patients were included) and qualitative study design. This limits the generalizability of these findings to the participants who were not interviewed and falls short of the aim to assess impact of NHIF and areas of improvement.	We thank the reviewer for the opportunity to clarify the issue of sample size in qualitative research. We agree that a clear sampling strategy is important, and we have now given additional detail about our strategy. In qualitative research we are seeking to understand a phenomenon, rather than to quantify it, and the approach to sampling depends on the nature of the research question, rather than a statistically proportionate probability sample necessarily being optimal (see Mason, J (2018) Sampling and selecting in qualitative research. Chapter 3 in Qualitative Researching (Third Edition), London: Sage, pp 53-82.) A purposive sampling strategy identifies a range of key categories relevant to the research question (here, for patients, urban/rural location, gender, age, condition/s) to ensure that the overall sample is balanced in relation to each category. This gives insight into the types of experiences reported by participants across these groups. Optimal sample size in qualitative studies is determined by saturation – a point where no new information emerges from additional data collection. Our sample for interviews (39) and FGDs (4) falls within what is considered sufficient for qualitative studies (9-17 for interviews and 4-8 for focus group discussions) based on a systematic review of literature to establish optimal sample sizes to achieve saturation in qualitative studies. (See, https://doi.org/10.1016/j.socscimed.2021.114523). We have now clarified this in the methods section. We agree that generally, the selection of two counties imposes a limit to generalizability, and we have acknowledged this in the limitation section. However, this is mitigated by the fact that our study focused on the role of the NHIF, which is a national entity that implements its programmes in a standard manner across the country. What is

	observed in one county is therefore very likely to be observed in other counties as well.
Moreover, authors' intent was to "generate evidence that would be relevant in informing policymakers and other stakeholders on how the configuration of health insurance reforms can be tailored to address the needs of NCD patients". This goal cannot reliably be achieved without an objective study where there are specific questions and responses to be picked by the respondents. For example, authors need to show policymakers whether implementation of NHIF increased the actual number of screening visits or follow up visits, if there is change in proportion of patients with controlled diabetes or blood pressures etc. If there is negative impact due to cost issues or lack of understanding, this can be captured by surveying the participants regarding the reasons for not following up or not taking the medications	We agree with the reviewer that quantitative measures to inform policy makers on necessary health insurance reforms is indeed valuable. In fact, this study is part of a larger study, which includes a quantitative assessment of the impact of NHIF using a matched cohort design. We think however, that a qualitative study that reports people's perceptions and experiences is also very important because it explains and complements quantitative findings. In this paper we report our qualitative findings. Our engagement with policy makers on the findings of the entire study will include findings from both the qualitative and quantitative work.
In summary, authors can use this qualitative data to build objective questionnaire to be circulated to eligible patients. This way they would have more participants and objective data as to the positive/negative impact of NHIF on the screening/control/treatment of NCDs in the population.	As mentioned before, we have indeed conducted a parallel quantitative study that examined whether NHIF protects households from incurring catastrophic health expenditures. Of note, the findings from both studies complement each other. The findings from this quantitative study have been submitted for publication.
The title and conclusion of the manuscript mentions "diabetes and hypertension" but majority of other text in the manuscript mentions "NCD". Even the responses from the respondents talk about issues with chemotherapy/radiotherapy/X-ray/PSA etc that are issues not closely related to diabetes/hypertension screening/management/follow up. This is again likely because respondents did not have a specific set of questions-answers to stick to which again makes it harder to apply the study findings to a broader population.	We thank the reviewer for this suggestion. We have edited the manuscript to specifically mention diabetes and hypertension only where relevant. The mentioning of other issues such as PSA by the one of the participants was meant to explain and give examples of other NCD tests that are not covered by the NHIF Supa Cover and hence are paid out-of-pocket by patients.

Authors conclude, “While the NHIF has reduced financial barriers and enhanced access to care for people living with hypertension and diabetes in Kenya, challenges persist that compromise NHIF’s responsiveness to the needs of these patients”. Again, this claim is only based on responses from few participants. What if an objective study later finds that majority patients believe that due to non-affordability of NHIF premiums, they actually have reduced access to care?	As mentioned above we do believe that a qualitative study is an important and original contribution to the evaluation of NHIF and as explained our sample size is considered adequate based on sample size requirements for qualitative studies. Further, as indicated earlier, we have indeed conducted an accompanying quantitative study, the findings of which complement this qualitative study. We have made this clearer in the discussion.
Introduction section is way too long before it gets to the main reasons of why this study was conducted. Consider it shortening.	We have now shortened the introduction section as suggested by the reviewer.
Authors report that NHIF coverage remains low at 16% but that data comes from a study in 2017. Do authors have more updated info on the current %age of population covered by NHIF?	We have followed the reviewer’s suggestion and included a more recent figure of 24% based on a recently published Kenya Demographic and Health Survey for 2022.
Study design and data collection, para 1: "FGD participants were selected from an existing PIC4-C scale-up study database". How were the participants selected. Based on eligibility criteria, there must be more eligible participants in the county. How many were screened for the study?	We used a purposive sampling approach (a sampling that is consistent with standard practice for qualitative studies) to select diabetic/hypertensive patients from the PIC4C database in 2021. Participants were sampled if they were enrolled into NHIF, were either diabetic/hypertensive, were above 18 years and consented to participate in the study. The PIC4C data were we purposively sampled the participants consisted of 769 households with at least one member living with either diabetes/hypertension or both.
Study design and data collection, para 2: "Data collection was discontinued when data saturation was attained". Can authors explain what they mean by that?	As suggested by the reviewer, we have now added an explanation in page 8 that data saturation in qualitative data collection means no new information is forthcoming during interviews/focus groups discussions.
Citation 23: The link lead to a website and not actual content/study/publication, so I am unable to verify if the data mentioned here is correct.	We apologize for the inconvenience; a new link has now been provided in reference 23.
Table 1: Need to cite the source of data for estimated hypertension and diabetes in respective counties.	Following the reviewer’s suggestions, we have now added citations for each source.
There are many incorrect usage of punctuations especially in the respondent answers but also in other parts of the manuscript. Since these were deduced from the interviews and not really written by the patients themselves, would recommend making sure sentences are coherent with correct punctuation usage.	We have followed the reviewer’s recommendations and the quotes provided in the manuscript have been edited where necessary.

VERSION 2 – REVIEW

REVIEWER	Chaturvedi, Abhishek Virginia Commonwealth University
REVIEW RETURNED	10-Apr-2023

GENERAL COMMENTS	1. The authors have fairly responded to the reviewer's comments. The biggest limitation at this point remains the "only qualitative" nature of the study. Perhaps adding the "quantitative findings" from the parallel study will further strengthen the manuscript and would be more interesting and informative to the readers of the journal. 2. I am surprised that the other reviewer could not even find one constructive criticism to improve upon this study/manuscript (whereas I was able to find many things to improve upon). Unfortunately, this gives me the impression that the other reviewer may have had some conflict of interest. Perhaps editors can identify another independent reviewer to make the peer review process unbiased. I am happy to re-review if the authors are willing to add objective data to the study. If not, then the manuscript may remain a weak one for publication in this journal (However, I would leave the final decision to the discretion of editors).
---

VERSION 2 – AUTHOR RESPONSE

Reviewer 2 Comments to Author	Response
1. The authors have fairly responded to the reviewer's comments. The biggest limitation at this point remains the "only qualitative" nature of the study. Perhaps adding the "quantitative findings" from the parallel study will further strengthen the manuscript and would be more interesting and informative to the readers of the journal.	We believe that there is value in providing qualitative findings, and that qualitative findings do not need to be backed by quantitative findings for them to have legitimacy. We also appreciate editorial comment that agrees with this position. The editorial comments confirm that the Journal does not expect reviewer 2's comments to be addressed in our revision.
2. I am surprised that the other reviewer could not even find one constructive criticism to improve upon this study/manuscript (whereas I was able to find many things to improve upon). Unfortunately, this gives me the impression that the other reviewer may have had some conflict of interest. Perhaps editors can identify another independent reviewer to make the peer review process unbiased.	We confirm that there are no pre-existing relationships with reviewer 1 known to any of the co-authors.